# Genome-Wide Identification and Analysis of *TCP* Gene Family among Three *Dendrobium* Species

**DOI:** 10.3390/plants12183201

**Published:** 2023-09-07

**Authors:** Yaoting Li, Lingli Li, Jiapeng Yang, Zhitao Niu, Wei Liu, Yi Lin, Qingyun Xue, Xiaoyu Ding

**Affiliations:** 1School of Life Sciences, Anhui Agricultural University, Hefei 230036, China; m17855262766@163.com (Y.L.); yjsc01@ahau.edu.cn (Y.L.); 2School of Biological and Pharmaceutical Engineering, West Anhui University, Lu’an 237012, China; 3College of Life Sciences, Nanjing Normal University, Nanjing 210023, China; m15895899536@163.com (L.L.); yangjiapengnj@163.com (J.Y.); niuzhitaonj@163.com (Z.N.); liuwei4@njnu.edu.cn (W.L.)

**Keywords:** *Dendrobium*, genome-wide analysis, expression profiles, gene family, TCP proteins

## Abstract

*Dendrobium* orchids, which are among the most well-known species of orchids, are appreciated for their aesthetic appeal across the globe. Furthermore, due to their strict living conditions, they have accumulated high levels of active ingredients, resulting not only in their medicinal value but also in their strong ability to respond to harsh environments. The *TCP* gene family plays an important role in plant growth and development, and signal transduction. However, these genes have not been systematically investigated in *Dendrobium* species. In this study, we detected a total of 24, 23, and 14 candidate *TCP* members in the genome sequences of *D. officinale*, *D. nobile*, and *D. chrysotoxum*, respectively. These genes were classified into three clades on the basis of a phylogenetic analysis. The *TCP* gene numbers among *Dendrobium* species were still highly variable due to the independent loss of genes in the CIN clade. However, only three gene duplication events were detected, with only one tandem duplication event (*DcTCP9*/*DcTCP10*) in *D. chrysotoxum* and two pairs of paralogous *DoTCP* gene duplication events (*DoTCP1*/*DoTCP23* and *DoTCP16*/*DoTCP24*) in *D. officinale*. A total of 25 *cis*-acting elements of *TCP*s related to hormone/stress and light responses were detected. Among them, the proportions of hormone response, light response, and stress response elements in *D. officinale* (100/421, 127/421, and 171/421) were similar to those in *D. nobile* (83/352, 87/352, and 161/352). Using qRT-PCR to determine their expression patterns under MeJA treatment, four *DoTCP*s (*DoTCP2*, *DoTCP4*, *DoTCP6*, and *DoTCP14*) were significantly upregulated under MeJA treatment, which indicates that *TCP* genes may play important roles in responding to stress. Under ABA treatment, seven *DoTCP*s (*DoTCP3*, *DoTCP7*, *DoTCP9*, *DoTCP11*, *DoTCP14*, *DoTCP15*, and *DoTCP21*) were significantly upregulated, indicating that *TCP* genes may also play an important role in hormone response. Therefore, these results can provide useful information for studying the evolution and function of *TCP* genes in *Dendrobium* species.

## 1. Introduction

Transcription factors (TFs) specifically bind to the promoter region of eukaryotic genes, and they play important roles in regulating the transcriptional initiation of specific sequences, which are fundamental to both plant development and responses to external environment stimulations [1,2]. TCP transcription factors, named after the four founding members, TEOSINTE BRANCHED1 (TB1) from *Zea mays* L., CYCLOIDEA (CYC) from *Antirrhinum majus* L., and PROLIFERATING CELL NUCLEAR ANTIGEN FACTORS 1 and 2 (PCF1 and PCF2) from *Oryza sativa* L. [3,4,5], feature the TCP domain, a highly conserved 59-residue-long basic helix–loop–helix (bHLH) structure at the N-terminus, which is associated with DNA binding, protein–protein interaction, and protein nuclear localization [6]. TCP proteins are a small family of plant-specific transcription factors and play important roles in multiple processes of plant growth and development by regulating cell growth and proliferation [7,8,9]. For example, in *Arabidopsis thaliana*, *AtTCP4* regulates jasmonic acid (JA) biosynthesis via the interaction with the JGB to further mediate pollen germination and gametophyte development. *TCP14* and *TCP15*, both members of TCP class I, were shown to function in cell proliferation and in several plant hormone signaling pathways.

According to the sequences of TCP conserved domains and phylogenetic relationships, members of the *TCP* gene family can be divided into two subfamilies: clade I and clade II. Clade II is also known as the PCF subfamily, while clade I TCP members are further divided into the CINCINNATA (CIN) and CYC/TB1 subfamilies [8,10]. The most obvious difference between these two subfamilies is that the basic region of the TCP domain of the clade I subfamily has four amino acids more than that of the clade II subfamily. In addition, several members of clade I have another conserved region outside the bHLH domains, named R domain, which is an arginine-rich motif containing 18–20 residues [8,11]. For instance, *AtTCP1* and *OsTB1* were identified due to the R domain in *Arabidopsis thaliana* and *Oryza sativa*, respectively.

The *TCP* gene family can participate in different processes of plant development, such as seed germination [12]; cell proliferation [13,14]; and leaf [15,16], flower [17], axillary bud [18], lateral branching [19], and pollen development [20]. For example, Tatematsu et al., using microarray and reporter gene analyses, indicated that *AtTCP14* regulates the activation of embryonic growth potential in *Arabidopsis thaliana* seeds. Takeda et al., by constructing transgenic *Arabidopsis thaliana* plants and performing reporter gene analysis, revealed that *TCP16* plays a crucial role in early processes in pollen development. In addition, the *TCP* gene family plays an important role in the response to various abiotic stresses, such as salt stress [21,22], drought stress [1,23], low temperature, and short photoperiod [24]. The *TCP* gene family also influences developmental and abiotic stress signaling via hormone pathways [25,26].

*Dendrobium*, one of the largest genera in Orchidaceae, includes approximately 1800 species, mainly distributed in tropical Asia, Australasia, and Australia [27]. There are about 120 species in China, and many of them are of great horticultural and medicinal value [27,28]. For example, both *Dendrobium officinale* Kimura et Migo and *Dendrobium nobile* Lindl. are famous traditional Chinese medicines (TCMs) due to their rich content of active ingredients, such as polysaccharides, flavonoids, and alkaloids [7,29], resulting in their excellent medicinal merits [27]. *Dendrobium chrysotoxum* Lindl. is popular for its high medicinal value and aesthetic appeal, while it can also be used in traditional Chinese food. With the rapid development of sequencing technology, the chromosomal-level genomes of these three species (*D. officinale*, *D. nobile*, and *D. chrysotoxum*) have been assembled with high quality [27,30,31], making it possible for researchers to explore the evolution of the *Dendrobium TCP* gene family.

In this study, *TCP* genes were identified in *D. officinale*, *D. nobile*, and *D. chrysotoxum*, and their chromosomal positions, structure features, conserved motifs, and *cis*-acting elements were analyzed. Phylogenetic analysis, collinearity analysis of the *TCP* genes, and prediction of their protein–protein interaction (PPI) networks were conducted. Lastly, on the basis of the *D. officinale* genome, tissue-specific expression analysis and transcriptome analysis were performed on the coding genes under MeJA treatment.

## 2. Results

### 2.1. Identification and Chromosomal Location of the TCP Genes in D. officinale, D. nobile, and D. chrysotoxum

To identify *TCP* family genes in *D. officinale*, *D. nobile*, and *D. chrysotoxum*, we used the BLAST method and hidden Markov model (HMM) to determine *TCP*s from their genome sequences, using 23 of the 24 known TCP protein sequences from *Arabidopsis* as references (Appendix A). A total of 24, 23, and 14 candidate *TCP* members were identified in *D. officinale*, *D. nobile*, and *D. chrysotoxum*, named *DoTCP1* to *DoTCP24*, *DnTCP1* to *DnTCP23*, and *DcTCP1* to *DcTCP14*, respectively (Appendix A). The *TCP* genes differed in their amino acid sequence length, isoelectric point (pI), and molecular weight (MW) among the three *Dendrobium* species (Appendix A). For example, the amino acid sequence length ranged from 179 (*DoTCP12*) to 584 (*DoTCP3*) in *D. officinale*, while it ranged from 211 (*DnTCP13*) to 786 (*DnTCP15*) in *D. nobile* and from 200 (*DcTCP8*) to 629 (*DcTCP1*) in *D. chrysotoxum*. The isoelectric point (pI) ranged from 5.82 (*DoTCP10*) to 11.28 (*DoTCP7*) in *D. officinale*, while it ranged from 6.16 (*DnTCP23*) to 10.39 (*DnTCP2*) in *D. nobile* and from 6.16 (*DcTCP6*) to 10.39 (*DcTCP1*) in *D. chrysotoxum*. The protein molecular weight (MW) ranged from 20.02 kDa (*DoTCP12*) to 64.87 kDa (*DoTCP3*) in *D. officinale,* while it ranged from 22.32 kDa (*DnTCP23*) to 63.93 kDa (*DnTCP2*) in *D. nobile* and from 21.19 kDa (*DcTCP8*) to 70.26 kDa (*DcTCP1*) in *D. chrysotoxum* (Appendix A).

The density and distribution of *TCP*s on chromosomes are shown in Figure 1. All 23 candidate *DnTCP*s in *D. nobile* were anchored on the chromosomes. On the other hand, 22 of the 24 were located on chromosomes of *D. officinale*, with *DoTCP23* and *DoTCP24* being instead located in unmapped regions (Figure 1A). Consistently, most *DcTCP*s were located on chromosomes of *D. chrysotoxum*, with only *DcTCP14* being located in an unidentified region (Figure 1B,C). In *D. officinale*, chromosome 6 contained the most *TCP* genes, followed by chromosomes 2, 11, 14, and 15, which contained two *TCP* genes, and chromosomes 1, 5, 7, 8, 9, 16, 17, and 19, with only one *TCP* gene (Appendix A; Figure 1A). In *D. nobile,* chromosomes 9, 13, and 17, contained three *TCP* genes, followed by chromosomes 5, 12, 14, and 18, which contained two *TCP* genes, and chromosomes 1, 2, 4, 6, 10, and 19, with only one *TCP* gene (Appendix A; Figure 1B). In *D. chrysotoxum*, chromosome 14 contained the most *TCP* genes, followed by chromosomes 5 and 12, which contained two *TCP* genes, and chromosomes 4, 6, 10, 13, 16, and 18, with only one *TCP* gene (Appendix A; Figure 1C).

### 2.2. Phylogenetic Relationship Analysis of TCP Gene Family

To explore the relationships among *Dendrobium TCP* genes, we conducted a phylogenetic analysis on 84 amino acid sequences, including 24 genes from *D. officinale*, 23 genes from *D. nobile*, 14 genes from *D. chrysotoxum*, and 23 genes from *A. thaliana* (Figure 2). The results inferred from the maximum likelihood (ML) evolutionary tree showed that the *TCP* genes were classified into three main classical subfamilies (class Ⅰ, class Ⅱ, and class Ⅲ). Among them, 28 genes (9 *DoTCP*s, 10 *DnTCP*s, and 9 *DcTCP*s) belonged to class I (PCF) (Figure 2). Class II and class Ⅲ represented CYC/TB1 (4 *DoTCP*s, 1 *DnTCP*, and 2 *DcTCP*s) and CIN (11 *DoTCP*s, 12 *DnTCP*s, and 3 *DcTCP*s), respectively (Figure 2).

### 2.3. Gene Structure Analysis and Conserved Motifs of TCPs

To better understand the *TCP* family genes in the three species of *Dendrobium*, we constructed three new evolutionary trees using the amino acid sequences of *DoTCP*, *DnTCP*, and *DcTCP*, respectively. Then, we analyzed the gene structure and conserved motifs of *TCP*s. Each of these three evolutionary trees of *TCP*s was divided into three branches (Figure 3). According to the CDS/UTR structure of each gene from the genome annotation files, we found that the gene structures of *TCP* genes were conserved (Figure 3). For example, the CDS of most *TCP* genes only had one exon, with only three genes containing one or two introns in each *Dendrobium* species (*DnTCP6* and *DnTCP22* with two introns and *DnTCP12* with one intron in *D. nobile*; *DcTCP1* with two introns and *DcTCP 6* and *DcTCP12* with one intron in *D. chrysotoxum*) (Figure 3).

According to the MEME online program, 15 conserved motifs were identified for each *Dendrobium* species, while their motif content and distribution were diversified (Figure 3). For example, in *D. officinale*, Motif 1, Motif 2, and Motif 3 were widespread on almost all *DoTCP* genes (Figure 3A), while Motif 9 and Motif 12 were specific to class Ⅰ (Figure 3A). Motif 5 was specific to class Ⅲ (Figure 3A). In *D. nobile*, Motif 1 was widespread on almost all *DnTCP* genes (Figure 3B), while Motif 5 and Motif 9 were specific to class Ⅲ (Figure 3B). In *D. chrysotoxum*, Motif 1 was widespread on almost all *DcTCP* genes (Figure 3C), while Motif 5 and Motif 9 were specific to class Ⅰ (Figure 3C).

### 2.4. Gene Collinearity Analysis

To further explore the expansion and evolution of *TCP* gene families, we analyzed duplication events of *TCP* genes. However, the evolution of the *TCP* gene family was conserved, with only one tandem duplication event (*DcTCP9*/*DcTCP10*) being found in *D. chrysotoxum* (Figure 4B) and two pairs of paralogous *DoTCP* gene duplication events (*DoTCP1*/*DoTCP23* and *DoTCP16*/*DoTCP24*) being identified in *D. officinale* (Figure 4A).

Collinearity analyses of the *TCP*s between *D. officinale* and the other plant species were used to explore evolutionary relationships. Notably, only three pairs of collinear gene pairs were found in the genomes of *D. officinale* and *A. thaliana* (Figure 5E). Collinear gene pairs of the *TCP*s between *D. officinale* and *A. thaliana* only existed on Chr 1 and Chr 5 of *A. thaliana* (Figure 5E). There were 11 pairs of collinear gene pairs of *TCP*s between *D. officinale* and *O. sativa*, which only existed on Chr 1, Chr 2, Chr 3, and Chr 12 (Figure 5D). There were 14 collinear gene pairs of *TCP*s between *D. officinale* and *V. planifolia*, and 15 pairs between *D. officinale* and *D. chrysotoxum* (Figure 5B,C). *D. nobile* was more closely related to *D. officinale*, with 26 pairs of collinear gene pairs (Figure 5A). The results showed that the *TCP* genes of *D. officinale* and *D. nobile* were highly conserved across their evolution history.

### 2.5. cis-Acting Element Analysis on Promotors of TCPs

To further explore the potential function of *TCP*s, we classified and identified *cis*-acting elements in the promoter regions. *cis*-Acting elements are closely related to gene expression patterns and functions. After removing the nonfunctional items, a total of 61 genes representing the remaining *cis*-acting elements of *TCP*s were divided into four categories, which were related to hormone response (ABRE, TCA-element, ERE, GARE-motif, P-box, TATC-box, TGACG-motif, TGA-element, etc.), light response (G-box, GATA-motif, 3-AF1-binding site, i-box, TCT-motif, AE-box, GT1-motif, etc.), stress response (Box 4, TC-rich repeats, MBS, Myc, Myb, S box, W box, etc.), and others (LTR, CAT-box, WUN-motif, etc.) (Appendix A; Figure 6).

In *D. chrysotoxum*, most *cis*-acting elements were related to light response (79/159), followed by hormone response (57/159) and stress response (10/159) (Figure 6C). The proportions of hormone response, light response, and stress response in *D. officinale* (100/421, 127/421, and 171/421) were similar to those in *D. nobile* (83/352, 87/352, and 161/352) (Figure 6A,B). Among the *cis*-acting elements of two *Dendrobium* species (*D. officinale* and *D. nobile*), the *cis*-acting elements related to stress response were the most abundant, followed by light response and hormone response (Figure 6A,B).

### 2.6. Expression of DoTCPs in Different Tissues and Expression Analysis under MeJA and ABA Treatments

To further study the function of *TCP*s, we analyzed the specific expression of *TCP*s in different tissues. Among the 24 *TCP* genes of *D. officinale*, 8 *DoTCP*s were obviously expressed (*DoTCP6*, *DoTCP9*, *DoTCP11*, *DoTCP13*, *DoTCP14*, *DoTCP15*, *DoTCP19*, and *DoTCP21*) (Figure 7A). According to the expression of these genes in different tissues, most genes were highly expressed in stems, flowers, and leaves but hardly expressed in roots (Figure 7A).

Since *cis*-acting element analysis showed that most *TCP*s in *D. officinale* corresponded to the stress response category, we used qRT-PCR to determine their expression patterns under MeJA treatment. Among them, eight *DoTCP*s (*DoTCP1*, *DoTCP3*, *DoTCP11*, *DoTCP13*, *DoTCP18*, *DoTCP19*, *DoTCP21*, and *DoTCP23*) were significantly downregulated, and four *DoTCP*s (*DoTCP2*, *DoTCP4*, *DoTCP6*, and *DoTCP14*) were significantly upregulated under MeJA treatment; the other genes showed no significant changes (Figure 7B).

In *DoTCP*s, there were more *cis*-acting elements of hormone response (ABRE); thus, we used qRT-PCR to determine their expression patterns under ABA treatment. Among them, six *DoTCP*s (*DoTCP1*, *DoTCP5*, *DoTCP8*, *DoTCP17*, *DoTCP20*, and *DoTCP22*) were significantly downregulated, and seven *DoTCP*s (*DoTCP3*, *DoTCP7*, *DoTCP9*, *DoTCP11*, *DoTCP14*, *DoTCP15*, and *DoTCP21*) were significantly upregulated; the other genes showed no significant changes (Figure 7C).

### 2.7. Protein–Protein Interaction Network of TCP Proteins in D. officinale

To further understand the function of genes, we analyzed the PPI network and detected the interaction between TCP proteins and related proteins using the STRING website.

A total of 72 proteins and 622 connections were identified, with TCP8 interacting with other proteins at least 38 times. Among these proteins, we found that (i) three genes (DoTCP5, DoTCP6, and DoTCP13) were related to TCP20, (ii) three proteins (DoTCP8, DoTCP17, and DoTCP21) were related to TCP2, and (iii) two proteins (DoTCP1 and DoTCP24) were related to TCP3 (Figure 8). These results indicated that TCP proteins of *D. officinale* have severe vital roles in multiple functions (Appendix A; Figure 8).

## 3. Discussion

### 3.1. The Loss of CIN Clade Genes Resulted in a Diversified TCP Gene Family among Dendrobium Species

*TCP* genes are well documented to regulate various physiological processes in plants, from hormone pathways [32,33], growth, and development [16,18,19], to the regulation of the circadian clock and defense [5,34]. For example, in *A. thaliana*, *AtTCP23* regulates plant development and flowering time [35], and *AtTCP14* regulates embryonic growth during seed germination [12]. With more and more plant genomes being published, numerous *TCP* gene families have been identified from many plants, with important roles in various physiological processes, especially related to their critical function in regulating plant growth and development. However, there remains a lack of information on the evolution and genome duplication of *TCP* genes in *Dendrobium* species.

In this study, a total of 24, 23, and 14 *TCP* members were identified in *D. officinale*, *D. nobile*, and *D. chrysotoxum*, respectively (Appendix A; Figure 1). Inferred from a previous study, the *TCP* genes were highly conserved in terms of their gene number and gene sequence. For example, the gene number was highly conserved. The number of *TCP* genes ranges from 6 *TCP* genes in *Physcomitrella patens* [36] to 74 in *Gossypium raimondii* [37]. However, in angiosperms, there are about 20 *TCP* members. The evolution of the *TCP* gene sequence was conserved. Consistent with previous studies, the *TCP* genes of *Dendrobium* species also belonged to two clades according to our phylogenetic analysis (Figure 2). Gene duplication is considered important in the variation of gene families, which are mainly classified as tandem duplication and segmental duplication according to the duplication patterns. Although only three gene duplication events happened among *Dendrobium* species, the *TCP* gene numbers among *Dendrobium* species were still highly variable, ranging from 14 *TCP* genes in *D. chrysotoxum* to 24 *TCP* genes in *D. officinale* (Appendix A; Figure 1). Inferred from our evolutionary tree of *TCP* genes, a total of 14 genes remained in *D. chrysotoxum* with only three genes in the CIN clade, indicating an independent loss of CIN genes in *D. chrysotoxum*.

Increasing studies have shown that CIN genes play important roles in the regulation of growth and development in plants, especially in lateral organ development. For example, in *A. thaliana*, *TCP* transcription factors negatively regulate the expression of specific genes to control stem side-organ morphology [38]. In tomato, miR319 regulates *LANCEOLATE*, which affects the development of composite leaves [39]. Compared with *D. officinale* and *D. nobile*, *D. chrysotoxum* has distinct characters in the habitat environment and photosynthesis pathway. For example, *D. officinale* and *D. nobile* are widely distributed in the southeastern part of China, while *D. chrysotoxum* is only distributed in Yunnan province in China [40]. Compared with the diversified habitat of *D. officinale* and *D. nobile*, which have a varied environment, *D. chrysotoxum* species are mainly distributed in warm regions. There are different photosynthesis pathways among the three species, whereby *D. officinale* and *D. nobile* are facultative CAM plants, while *D. chrysotoxum* is a C3 plant [41]. Considering the diversified habitats, photosynthesis pathways, and the critical functions of CIN genes, we suggest that the loss of CIN genes in *D. chrysotoxum* led to a compact *TCP* gene family of *D. chrysotoxum*, resulting a diversified *TCP* gene family among *Dendrobium* species.

### 3.2. The Harsh Habitats and Different Photosynthesis Pathways Have Resulted in the Diversity of Biological Functions of the TCP Gene Family among Dendrobium Species

Previous studies have shown that TCP proteins play a critical role in molecular mechanisms such as plant organ development, hormone signaling regulation, and defense against abiotic stress [42,43]. However, different genes in the *TCP* gene family have different functions, with the same gene in different species sometimes having different functions. For instance, the *cis*-acting element analysis showed that some *TCP* genes respond to hormones, light, or stress. Both *DcTCP12* and *DoTCP24* were subclaves of CYC; *DcTCP12* responded to stress, whereas *DoTCP24* did not (Figure 2 and Figure 6).

In this study, on the basis of our comparative analysis, we believe that the harsh habitats and different photosynthesis pathways resulted in the diversity of biological functions of the *TCP* gene family among *Dendrobium* species.

Firstly, a total of 25 *cis*-acting elements of *TCP*s related to hormone/stress and light responses were detected. Among them, 18 genes were related to hormone response, and 22 genes were related to light response, while 36 genes were related to stress response (Figure 6).

Secondly, the diverse gene expression patterns of *TCP* genes indicated that *TCP* genes were mainly a function of the harsh habitats and different photosynthesis pathways. For example, the gene expression under MeJA treatment revealed four upregulated genes related to defense against abiotic stress (Figure 7). Only *DoTCP6* and *DoTCP14* showed high gene expression levels in stems, leaves, and flowers (Figure 7). *Dendrobium* species grow under strict conditions, such as epiphytic environments on cliffs or tree trunks, or distributed at high altitudes, above 1200 m, leading to the accumulation of high content of active ingredients, i.e., polysaccharides, flavonoids, and alkaloids, in their stems. The high content of polysaccharides, flavonoids, and alkaloids also improves the stress response to harsh habitats, resulting in a higher gene expression of *TCP* genes than other organisms (Figure 7A). A total of five genes related to light stress were expressed in leaves (Figure 7A). *Dendrobium* species are diversified in their photosynthesis pathways; for instance, *D. chrysotoxum*, *Dendrobium Chrysanthum* Lindl., *Dendrobium Lindleyi* Stendel, *Dendrobium Longicornu Lindl.*, and *Dendrobium thyrsiflorum* Rchb. f. are C3 plants; *Dendrobium terminale* Par. et Rchb. f., *Dendrobium acinaciforme* Roxb., and *Dendrobium primulinum* Lindl. are crassulacean acid metabolism (CAM) plants; *D. nobile*, *D. officinale*, and *Dendrobium hercoglossum* Rchb. f. are recognized as C3-CAM plants [41]. The higher expression of *DoTCP* genes may improve the response of *D. officinale* to light stress by changing the photosynthesis pathway from C3 to CAM upon exposure to light stress. Additionally, there were seven genes (*DoTCP6*, *DoTCP9*, *DoTCP11*, *DoTCP13*, *DoTCP14*, *DoTCP15*, and *DoTCP21*) expressed in the flowers of *DoTCP*s (Figure 7A), potentially related to hormone response. Seven *DoTCP*s (*DoTCP3*, *DoTCP7*, *DoTCP9*, *DoTCP11*, *DoTCP14*, *DoTCP15*, and *DoTCP21*) were significantly upregulated under ABA treatment (Figure 7C), suggesting that these genes play an important role in hormone response.

Thirdly, the protein–protein interaction network analysis showed a close relationship among the DoTCP proteins, indicating that *TCP* genes play important roles in response to *PCF1*. For example, the genes with the most connections, such as *PCF1* and *PCF2*, were related to plant growth and development. Therefore, we speculate that *TCP* genes play a more important role in plant growth and development.

## 4. Materials and Methods

### 4.1. Identification of TCP Genes in D. officinale, D. nobile, and D. chrysotoxum Genome

The hidden Markov model (HMM) map of *TCP* (PF03634) was downloaded from the protein family database (Pfam) (http://pfam-legacy.xfam.org/ (accessed on 22 August 2022)), and candidate proteins with an E-value of 1 × 10^−5^ were identified using HMMER v3.2.1. The TCP protein sequences of *A. thaliana* were downloaded, and BLASTP was used to query the proteins of *D. officinale*, *D. nobile*, and *D. chrysotoxum*. The TCP protein sequences identified with the two methods were integrated. In order to exclude false-positive sequences, the Pfam, NCBI-CDD (https://www.ncbi.nlm.nih.gov/Structure/cdd/wrpsb.cgi (accessed on 22 August 2022)), and SMART (http://smart.embl-heidelberg.de/ (accessed on 22 August 2022)) databases that submitted the identified TCP protein sequences were selected for further analysis of the *TCP* gene family. The protein molecular weight (kDa), aliphatic index (AI), and theoretical isoelectric point (pI) were estimated using ExPASy software [44]. Lastly, the chromosome position was analyzed using TBtools v1.6 [45].

### 4.2. Phylogenetic Relationship

Multiple sequence alignments of *TCP* full-length amino acid sequences of *D. officinale*, *D. nobile*, *D. chrysotoxum*, and *A. thaliana* were compared using MAFFT v7.487 software [46], and the maximum likelihood evolutionary trees were constructed using RAxML v1.3 software with the CAT + GTR model and 1000 bootstrap replicates [47].

### 4.3. Motifs and Domains

Firstly, MEME (https://meme-suite.org/meme/ (accessed on 22 August 2022)) was used to determine conserved motifs, with the number of motifs set to 20 and all other parameters set to their default values. Then, the exon–intron structure of each sequence was displayed online using GSDS software (http://gsds.gao-lab.org/index.php (accessed on 22 August 2022)). Lastly, the characteristic motifs and domains of the *TCP* gene family were aligned using TBtools v1.6 [45].

### 4.4. Gene Duplication and Syntenic Analysis

DNA sequences of the *TCP* gene family of *D. officinale*, *D. nobile*, and *D. chrysotoxum* were compared using BLASTN with an e-value of 1 × 10^−20^. Gene duplication was identified using BLASTN results and analyzed using MCScanX, and the results were visualized using Circos. MCScanX (cscore ≥ 0.7) was employed to detect and display the syndesmosis between *D. officinale* and other plant genomes.

### 4.5. Promoter Analysis

The region 1500 bp upstream of the *TCP* gene translation initiation site was extracted as the presumed promoter and submitted to the PlantCare database online to analyze the assumed *cis*-acting elements (https://bioinformatics.psb.ugent.be/webtools/plantcare/html/, accessed on 22 July 2023). The characteristic *cis*-acting elements of the *TCP* gene family were aligned using TBtools v1.6 [45].

### 4.6. Expression of TCPs in Different Tissues

To analyze the expression pattern of the *TCP* genes in *D. officinale*, we searched the NCBI SRA database for RNA-sequencing data from root (SSR2014227 and SSR2014230), stem (SSR1917040, SSR1917041, SSR1917042, and SSR1917043), leaf (SSR2012297 and SSR2014325), and flower (SSR2012396 and SSR2014476) as four different tissues. The FPKM value of *DoTCP*s was calculated using StringTie v2.2.0 (http://string-db.org/, accessed on 22 July 2023) to estimate the abundance of transcripts, and the heat map was built using TBtools v1.6 to visualize the expression [45].

### 4.7. Quantitative Real-Time PCR

Firstly, total RNA was extracted using the EASY spin Plant RNA Kit (Aidlab, Beijing, China). Then, HiScript^®^ III-RT SuperMix for qPCR (Vazyme, Beijing, China) was used to synthesize the first-strand cDNAs. Snapgene software was used to design primers (Appendix A), and the ABI-7500 Connect Real-Time PCR Detection System was used to perform qRT-PCR. With 1 μL of template in a reaction volume of 20 μL, cDNAs were diluted to 200 ng, and three technical repetitions were performed. The PCR amplification procedure was as follows: 95 °C for 30 s, 95 °C for 10 s, 60 °C for 30 s, and 60 °C for 15 s. GAPDH was used as the internal reference gene. Primer sequences are presented in Appendix A.

### 4.8. Protein–Protein Interaction Network

The expression data were calculated using the 2^−∆∆CT^ method [48]. The STRING database was used to align the DoTCP protein sequences and predict the relationships. The regulatory networks were visualized using Cytoscape v3.7.2 software [49].

## 5. Conclusions

*T*his study explored the evolution of *TCP* genes in *D. officinale*, *D. nobile*, and *D. chrysotoxum* by analyzing the conserved motifs, gene structure, gene duplication, and collinearity. Gene motif and structure analysis showed that genes in the same clade had relatively conserved gene structures and similar protein motifs. The differentially expressed genes screened under MeJA and ABA treatment using qRT-PCR may play an important role in stress response and hormone response. On the basis of our comparative analysis, we suggest that the harsh habitats and different photosynthesis pathways resulted in the diversity of biological functions of the *TCP* gene family among *Dendrobium* species. This study lays a foundation for the genetic evolution and functional analysis of *TCP* genes in *Dendrobium*.

## Figures and Tables

**Figure 1 plants-12-03201-f001:**
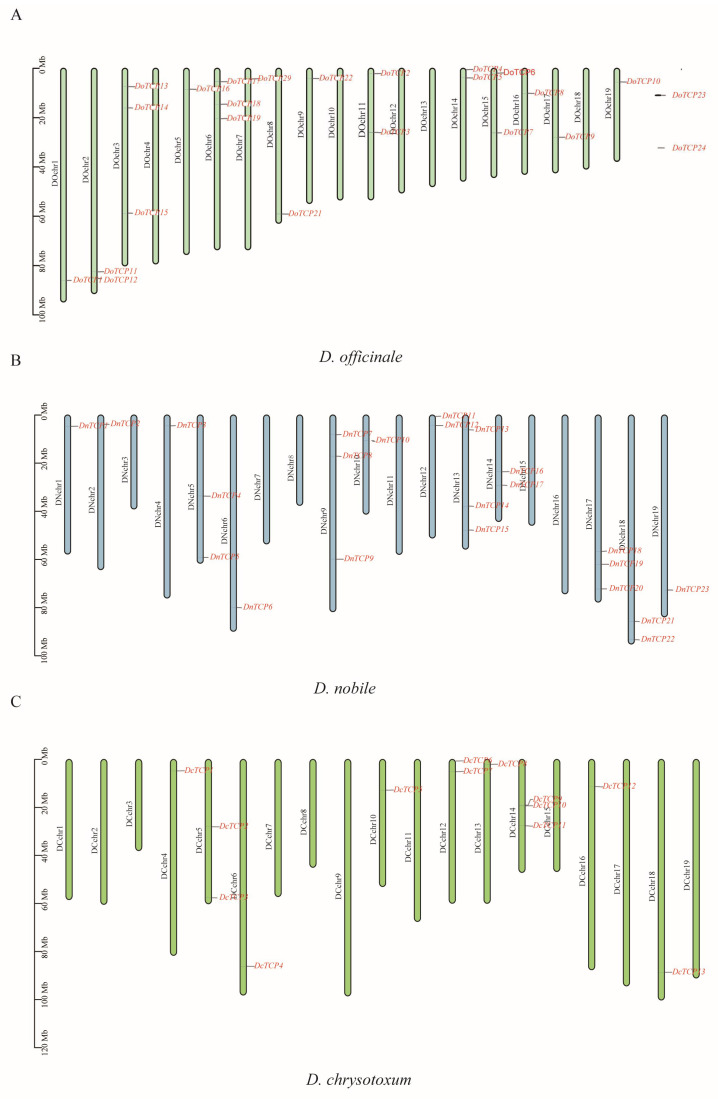
Chromosomal distributions of the identified *TCP* genes in *D. officinale* (**A**), *D. nobile* (**B**), and *D. chrysotoxum* (**C**). Green, blue, and yellow-green colors represent *D. officinale*, *D. nobile*, and *D. chrysotoxum*, respectively. The red letters represent the names of *TCP* genes.

**Figure 2 plants-12-03201-f002:**
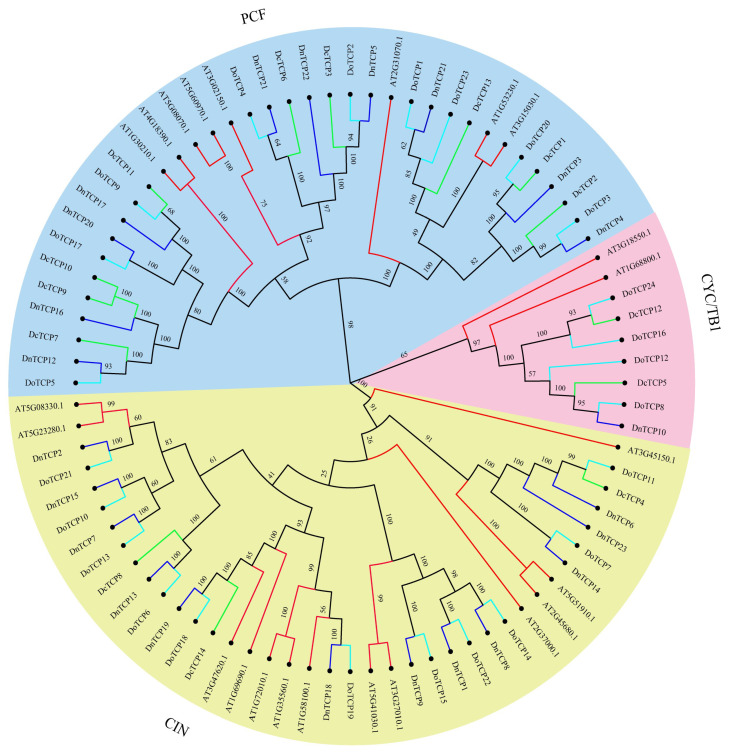
Phylogenetic relationships of *TCP*s in *D. officinale*, *D. nobile*, *D. chrysotoxum*, and *A. thaliana*. The maximum likelihood evolutionary tree was constructed with RAxML with 1000 bootstraps replicates. Light-blue, dark-blue, green, and red colors represent TCP protein sequences from *D. officinale* (Do), *D. nobile* (Dn), *D. chrysotoxum* (Dc), and *A. thaliana* (At), respectively. Different subfamilies are shaded with different colors.

**Figure 3 plants-12-03201-f003:**
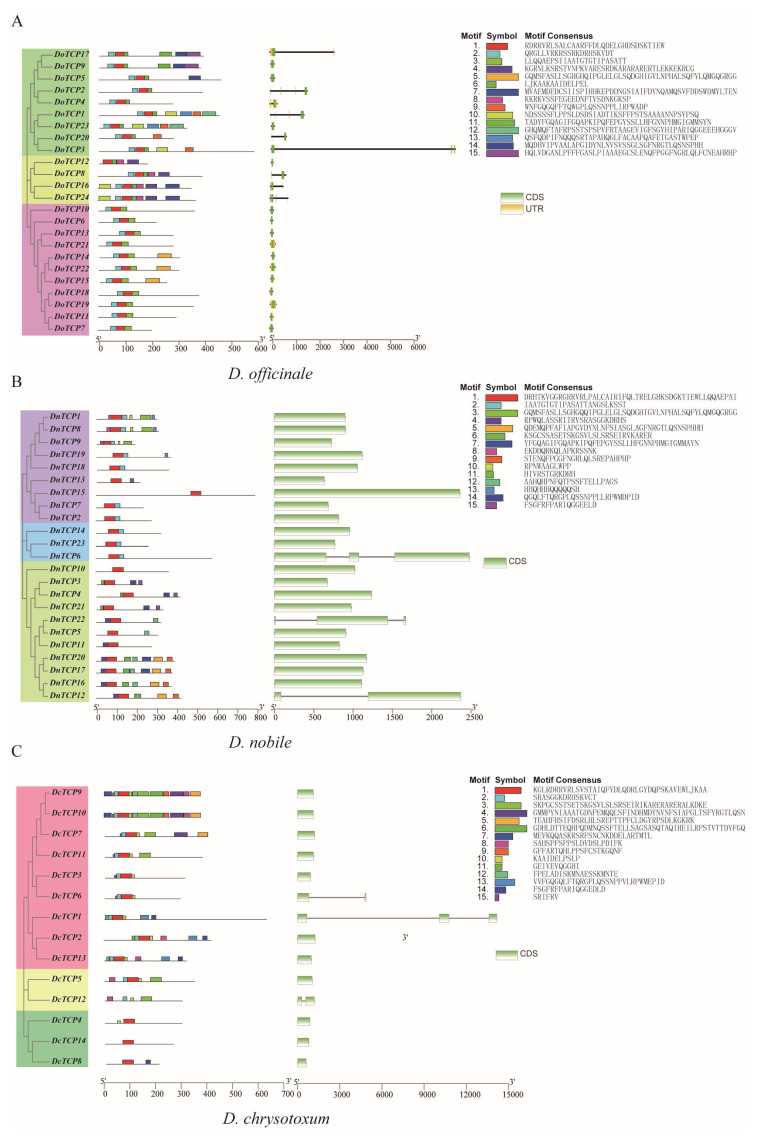
Phylogenetic relationships, conserved motifs, and exon–intron structures of *TCP* genes in *D. officinale* (**A**), *D. nobile* (**B**), and *D. chrysotoxum* (**C**). Different colors represent the 15 different motifs. Different branches are shaded with different colors. The black lines indicate introns.

**Figure 4 plants-12-03201-f004:**
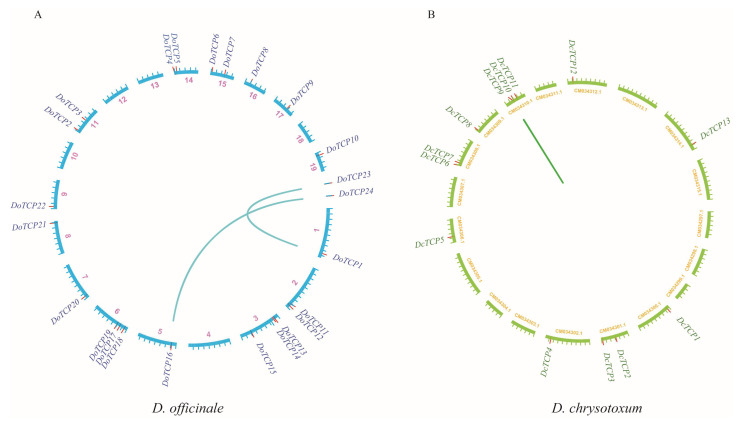
Schematic representations of the gene duplications of *TCP* genes from *D. officinale* (**A**) and *D. chrysotoxum* (**B**). Blue and green lines highlight the syntenic gene pairs in *D. officinale* (Do) and *D. chrysotoxum* (Dc), respectively.

**Figure 5 plants-12-03201-f005:**
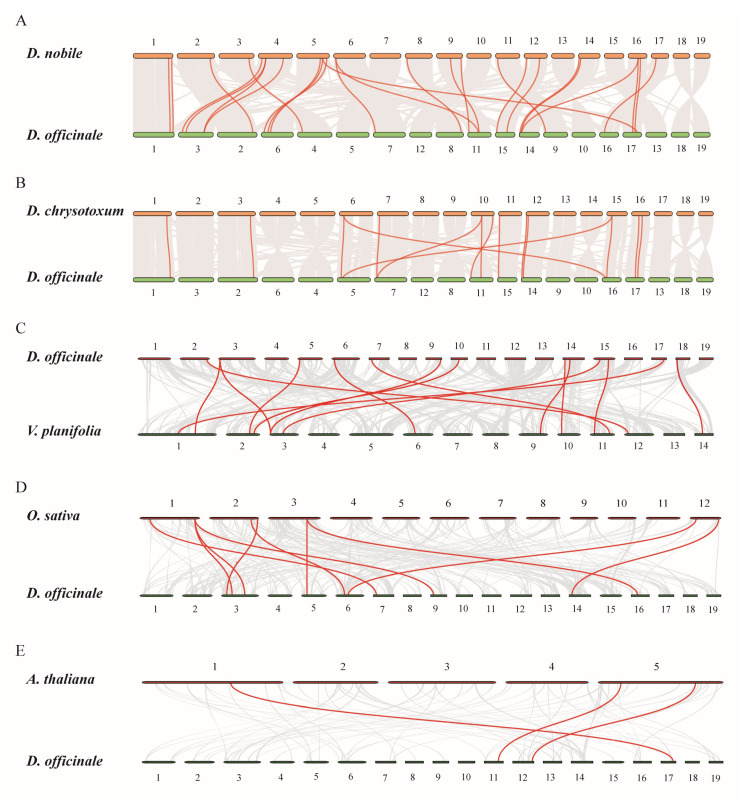
Collinearity analysis of *TCP* genes between *D. officinale* and five other plants, including *D. nobile* (**A**), *D. chrysotoxum* (**B**), *V. planifolia* (**C**), *O. sativa* (**D**), and *A. thaliana* (**E**). Gray lines indicate the collinear blocks. Syntenic genes of the *TCP* gene family are exhibited with red lines. The numbers in the figure represent the number of the corresponding chromosome.

**Figure 6 plants-12-03201-f006:**
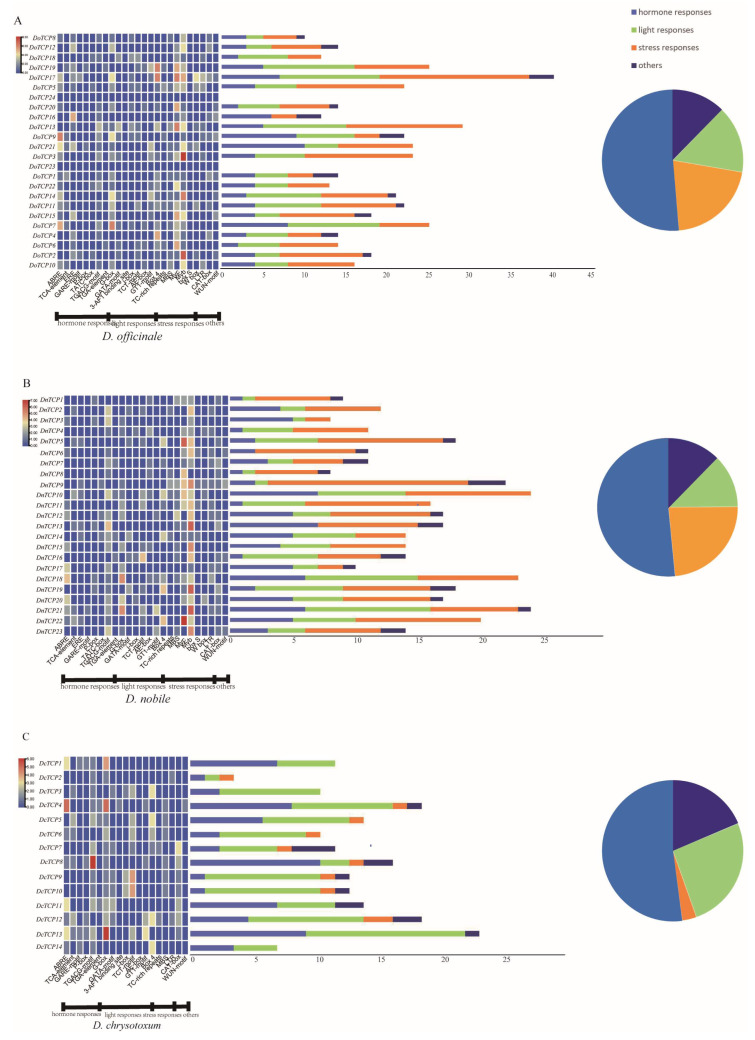
Information on *cis*-acting elements in *TCP* genes of *D. officinale* (**A**), *D. nobile* (**B**), and *D. chrysotoxum* (**C**). The different colors in the histogram indicate the number of *cis*-elements in each category.

**Figure 7 plants-12-03201-f007:**
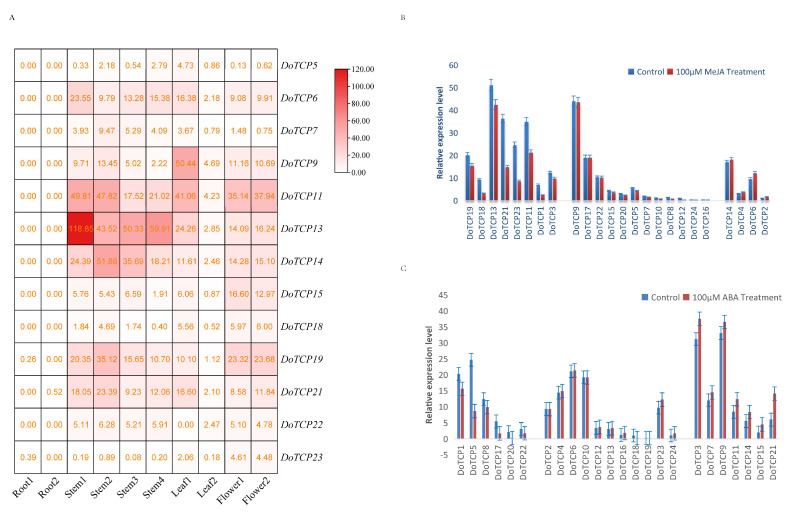
Expression analysis of *DoTCP*s in different tissues under MeJA treatment. (**A**) Expression profiles of *TCP* genes of *D. officinale* in different tissues including root, stem, leaf, and flower. Z-score-transformed FPKM values. (**B**) Relative expression levels of *DoTCP*s under MeJA treatments. (**C**) Relative expression levels of *DoTCP*s under ABA treatments.

**Figure 8 plants-12-03201-f008:**
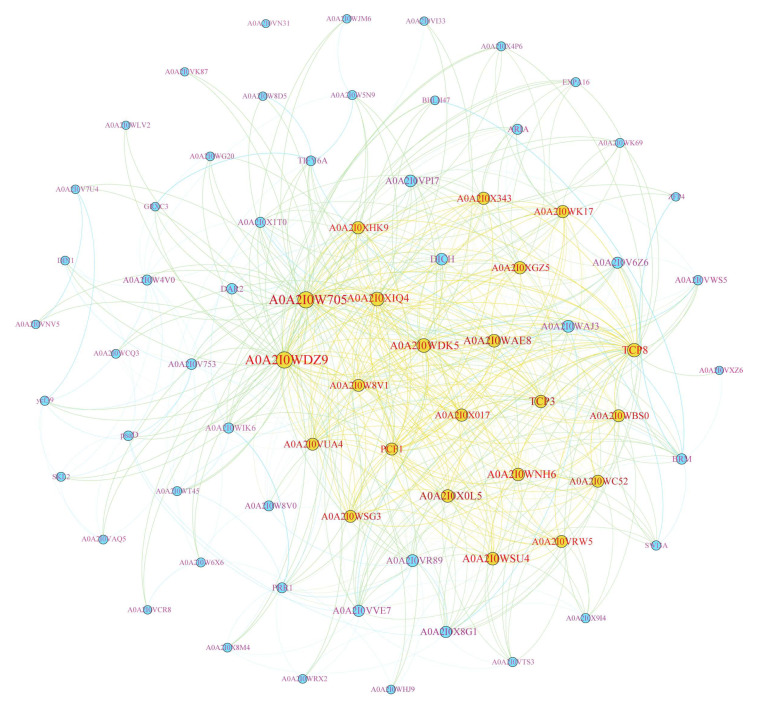
Protein–protein interaction (PPI) networks of TCP proteins in *D. officinale*. The gradient circle size and color indicates the degree of importance. The important of yellow to a greater extent than blue.

## Data Availability

All data generated or analyzed during this study are included in the article and Appendix A.

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
