# Peer review of "Genome-Wide Identification and Analysis of TCP Gene Family among Three Dendrobium Species"

_plants, 2023, doi:10.3390/plants12183201_

Round 1
Reviewer 1 Report (Previous Reviewer 2)
I could not any improvement from the earlier Ms. For example, fig 7A, I could still see 0 expression value for many genes in all the stages, which showed that authors dont have any idea of representing the results. What is the significance of showing them in a heatmap? Those could be removed. Further, I could not see any elaboration on the stages in the figure legends.
What is root1, root2 etc.?
Need revision.
Author Response
Dear editor and reviewers,
On behalf of my co-authors, we thank you very much for giving us an opportunity to revise our manuscript, we appreciate the editor and reviewers very much for their positive and constructive comments and suggestions. Those comments are all valuable and very helpful for revising and improving our manuscript. We have studied the comments carefully and tried our best to revise our manuscript based on the comments, which we hope meets with approval.
Response to Reviewer 1:
- I could not any improvement from the earlier Ms. For example, fig 7A, I could still see 0 expression value for many genes in all the stages, which showed that authors dont have any idea of representing the results. What is the significance of showing them in a heatmap?Those could be removed.
Response:
Many thanks for your valuable and helpful suggestion.
For Figure 7A, we remove the gene with expression value 0 from the heat map and redraw Figure 7A.
- Further, I could not see any elaboration on the stages in the figure legends.
Response:
Many thanks for your valuable and helpful suggestion.
Figure 7A is the FPKM value of Dendrobium officinale TCP gene expression profile in different tissues such as root, stem, leaf and flower, but not the expression quantity of TCP gene at each stage.
- What is root1, root2 etc.?
Response:
Many thanks for your valuable and helpful suggestion.
To analyze the expression pattern of the TCP genes in D. officinale, we searched the NCBI SRA database for RNA-sequencing data from the root (SSR2014227 and SSR2014230), stem (SSR1917040, SSR1917041, SSR1917042, and SSR1917043), leaf (SSR2012297 and SSR2014325), and flower (SSR2012396 and SSR2014476) as four different tissues. Root1 and root2 represent the FPKM values of TCP genes in SSR2014227 and SSR2014230, respectively.
Reviewer 2 Report (Previous Reviewer 1)
The authors have fulfilled all the requirements according the comments/ suggestions. The paper is very suitable for publication in this journal and very useful for the scientific community.
Author Response
Dear editor and reviewers,
On behalf of my co-authors, we thank you very much for giving us an opportunity to revise our manuscript, we appreciate the editor and reviewers very much for their positive and constructive comments and suggestions. Those comments are all valuable and very helpful for revising and improving our manuscript. We have studied the comments carefully and tried our best to revise our manuscript based on the comments, which we hope meets with approval.
Response to Reviewer 2:
The authors have fulfilled all the requirements according the comments/ suggestions. The paper is very suitable for publication in this journal and very useful for the scientific community.
Response:
Many thanks for your recognition of the author's modification of this paper and the content of this paper.
Round 2
Reviewer 1 Report (Previous Reviewer 2)
NA
NA
Author Response
Dear editor and reviewers,
On behalf of my co-authors, we sincerely thank you for giving us the opportunity to revise our manuscript. We greatly appreciate the editor and reviewers for their positive and constructive comments and suggestions. These comments are invaluable and incredibly helpful for revising and improving our manuscript. We have carefully studied the comments and put forth our best effort to revise the manuscript in accordance with them. We hope that these revisions meet with your approval.
Thank you once again for your time and consideration.
Response to reviewer 1:
- Extensive editing of English language required.
Response:
Many thanks for your valuable and helpful suggestion.
The English version of this manuscript has been thoroughly revised by the professional English editorial team of MDPI Magazine.
This manuscript is a resubmission of an earlier submission. The following is a list of the peer review reports and author responses from that submission.
Round 1
Reviewer 1 Report
Review Report
Manuscript ID: plants-2428988
The manuscript entitled “Genome-wide Identification and Analysis of TCP Gene Families among Three Dendrobium Species” submitted to plants needs corrections and improvements. Generally, the manuscript is fine but still there are some minor comments for the improvement of the manuscript.
Abstract: The abstract needs to be precise, informative, and to the point. Please mention the specific objective of your study
Introduction: In the introduction part please mention the history, importance and background of Dendrobium orchid and their use in molecular research and rearrange the paragraph accordingly.
Results: Figure 1 regarding Chromosomal distributions of the identified TCP genes in D. officinale, D. nobile, and D. 126 chrysotoxum is not very clear. Please revise the figure 1 with more clear bars/figures.
Figure 3 is not very and readable. Please revise the Figure 3 regarding Phylogenetic relationships, conserved motifs and exon-intron structures of TCP genes in 166 D. officinale (A), D. nobile (B), and D. chrysotoxum (C).
Please revise the figure 4. Schematic representations of the gene duplications of TCP genes from D. officinale (Do) 187 and D. chrysotoxum (Dc) is not very clear and readable.
Figure 5 is not very clear and readable. Please revise the figure 5. Collinearity analysis of TCP genes between D. officinale and five other plants, including D. 191 nobile, D. chrysotoxum, O. sativa, V. planifolia and A. thaliana.
Figure 6 is not very clear and not readable. Information of cis-acting elements in TCP genes of D. officinale (A), D. nobile (B), and D. 214 chrysotoxum (C).
Discussion: Improve the discussion part with respect to genome evolution of Dendrobium Species.
Materials and Methods: This part is ok. Please add more information about identification analysis of dendrobium species.
Carefully check all spellings, numerical order of references, format following the requirements of the submitted journals.
Cross check all the references cited in the text with reference list

Quality of english is fine. Minor editing is required.
Author Response
Dear editor and reviewers,
On behalf of my co-authors, we thank you very much for giving us an opportunity to revise our manuscript, we appreciate editor and reviewers very much for their positive and constructive comments and suggestions. Those comments are all valuable and very helpful for revising and improving our manuscript. We have studied comments carefully and tried our best to revise our manuscript according to the comments, which we hope meet with approval.
Response to Reviewer 1:
- Abstract: The abstract needs to be precise, informative, and to the point. Please mention the specific objective of your study
Response:
Many thanks for your valuable and helpful suggestion.
The content of the abstract has been streamlined to make it more organized.
- Introduction: In the introduction part please mention the history, importance and background of Dendrobiumorchid and their use in molecular research and rearrange the paragraph accordingly.
Response:
Many thanks for your valuable and helpful suggestion.
The introduction to Dendrobium has been added and the paragraphs rearranged.
- Results: Figure 1 regarding Chromosomal distributions of the identified TCPgenes in officinale, D. nobile, and D. chrysotoxum is not very clear. Please revise the figure 1 with more clear bars/figures.
Figure 3 is not very and readable. Please revise the Figure 3 regarding Phylogenetic
relationships, conserved motifs and exon-intron structures of TCP genes in D. officinale (A), D. nobile (B), and D. chrysotoxum (C).
Please revise the figure 4. Schematic representations of the gene duplications of TCP genes from D. officinale (Do) and D. chrysotoxum (Dc) is not very clear and readable.
Figure 5 is not very clear and readable. Please revise the figure 5. Collinearity analysis of TCP genes between D. officinale and five other plants, including D. nobile, D. chrysotoxum, O. sativa, V. planifolia and A. thaliana.
Figure 6 is not very clear and not readable. Information of cis-acting elements in TCP genes of D. officinale (A), D. nobile (B), and D. chrysotoxum (C).
Response:
Many thanks for your valuable and helpful suggestion.
In order to improve the clarity of the pictures, the text in the pictures has been enlarged, and the clarity of the pictures has been improved, making the pictures easier to read. The Figure 3 regarding Phylogenetic relationships, conserved motifs and exon-intron structures of TCP genes in D. officinale (A), D. nobile (B), and D. chrysotoxum (C) have been modified.
- Discussion: Improve the discussion part with respect to genome evolution of Dendrobium
Response:
Many thanks for your valuable and helpful suggestion.
In Discussion 1, the evolution of TCP genes in Dendrobium was discussed with these three Dendrobium species (D. officinale, D. nobile, and D. chrysotoxum)as research objects, and an important point was grasped: TCP genes in CIN branch was lost in D. chrysotoxum, and the reason for this phenomenon was analyzed.
- Materials and Methods: This part is ok. Please add more information about identification analysis of dendrobium
Carefully check all spellings, numerical order of references, format following the requirements of the submitted journals.
Cross check all the references cited in the text with reference list
Response:
Many thanks for your valuable and helpful suggestion.
The format of this article, all spellings, and references have been checked, supplemented, and revised.
Reviewer 2 Report
In the Ms Genome-wide Identification and Analysis of TCP Gene Fami- 2 lies among Three Dendrobium Species, authors identified a total of 24, 23, 14 and 14 candidate TCP members in the genome sequences of D. officinale, D. nobile, and D. chrysotoxum, and used for in-silico characterization and expression analysis. The study is ok, but the Ms is very poorly written. The Ms needs to be first corrected for language then it can be scientifically reviewed.
For instance line 28, play important roles in responded to stress. ?
How many TCP genes are present in Arabidopsis? Why number is variable in different species of Dendrobium? It seems to be not identified properly.
Ms is horribly written.
4.2. Phylogenetic relationship and gene structure 365
“Use MAFFT v7.487 software to compare the full-length amino acid sequences of TCP 366 [45], and use RAxML v1.3 software with a bootstrap value of 1,000 to build the maximum 367 likelihood phylogenetic trees.”
Is it the way of writing? The methods should be completely revised. Authors may take the help for language correction.
I could not see any internal control in Rt PCR, usually two gene are now used as described www.mdpi.com/2075-1729/12/7/941
Authors should also include evolutionary analysis as described www.mdpi.com/2223-7747/11/7/911, because this study is mostly in-silico.
RT PCR methods and interaction methods should be separately written. Authors should also use stitch for interaction analysis. You may follow www.mdpi.com/2223-7747/11/5/587
Figure quality should be improved. Most of the written parts are invisible in the majority of figures.
In Figure 7A, I don’t see any expression value in most of the genes. How it is possible?
Discussion seems to be repetition of results in the many paragraphs. Authors may refer the suggested Ms during the revision of Ms. Only important findings could be discussed.
Conclusion should be rewritten with future perspective. It should not be the repeat of abstract.
Quality of English Language is very poor.
Author Response
Dear editor and reviewers,
On behalf of my co-authors, we thank you very much for giving us an opportunity to revise our manuscript, we appreciate editor and reviewers very much for their positive and constructive comments and suggestions. Those comments are all valuable and very helpful for revising and improving our manuscript. We have studied comments carefully and tried our best to revise our manuscript according to the comments, which we hope meet with approval.
Response to Reviewer 2:
- How many TCP genes are present in Arabidopsis? Why number is variable in different species of Dendrobium? It seems to be not identified properly.
Response:
Many thanks for your valuable and helpful suggestion.
The number of TCP genes in Arabidopsis Thaliana is 24, as shown in 2.1.(To identify TCP family genes in D. officinale, D. nobile, and D. chrysotoxum, we used BLAST method and hidden Markov model (HMM) to determine TCPs from their genome sequences using 24 known TCP protein sequences from Arabidopsis as references.)
The reasons why the number of different Dendrobium species is variable are presented in the discussion. The evolution of TCP genes in Dendrobium was discussed with these three Dendrobium species (D. officinale, D. nobile, and D. chrysotoxum)as research objects, and an important point was grasped: TCP genes in CIN branch was lost in D. chrysotoxum, and the reason for this phenomenon was analyzed.
- 2. Phylogenetic relationship and gene structure 365
“Use MAFFT v7.487 software to compare the full-length amino acid sequences of TCP 366 [45], and use RAxML v1.3 software with a bootstrap value of 1,000 to build the maximum 367 likelihood phylogenetic trees.”
Is it the way of writing? The methods should be completely revised. Authors may take the help for language correction.
Response:
Many thanks for your valuable and helpful suggestion.
The writing of material method has been modified to make the language more logical.
- Figure quality should be improved. Most of the written parts are invisible in the majority of figures.
In Figure 7A, I don’t see any expression value in most of the genes. How it is possible?
Response:
Many thanks for your valuable and helpful suggestion.
In order to improve the clarity of the pictures, the text in the pictures has been enlarged, and the clarity of the pictures has been improved, making the pictures easier to read.
- Discussion seems to be repetition of results in the many paragraphs. Authors may refer the suggested Ms during the revision of Ms. Only important findings could be discussed.
Response:
Many thanks for your valuable and helpful suggestion.
In the discussion, questions are raised and hypotheses are made based on the results of the research, and the reasons for such results are analyzed and discussed based on references.
- Conclusion should be rewritten with future perspective. It should not be the repeat of abstract.
Response:
Many thanks for your valuable and helpful suggestion.
The conclusion has been revised and written.
Reviewer 3 Report
Dear authors,
This article is concerning a research work entitled “Genome-wide Identification and Analysis of TCP Gene Families among Three Dendrobium Species by Yaoting Li, Lingli Li, Jiapeng Yang, Zhitao Niu, Wei Liu, Yi Lin, Qingyun Xue and Xiaoyu Ding. As interesting data, I recommend it for an international audience in this journal, however several points have to be considered by the authors, and a major revision is requested.
Please notice that in order to bring a broad audience to this paper and to this journal, for specialists and non-specialists, the six major points of my comments (at the beginning) are very important (mandatory…) for a suitable value of the article. Minor points are also enhanced at the end of this review.
I deeply hope to see this good article published soon,
The six major points are:
11- The first important point is that it is very ambiguous to read this manuscript, as we do not know exactly if results correspond to database or to laboratory experiment (your point 4.7? For instance in 2.6, "were obviously expressed" means with database or experimentally? See also in 3.1 “members were identified”; in 4.5, do you mean that you experimented for "The genomic DNA sequences of 1500 bp upstream of TCP genes were "extracted"...", if so how did you get the plants and how did you cultivate them and where does it appears in material and methods?
22- The second important point concerns the discussion and the (complex but very interesting) hypotheses enhanced, unfortunately too general and too weakly sustained because of 1/genes, 2/habitats and 3/relationship genes-proteins-habitats. 1/1 From how many entries (= number of plants analysed; from databases or others) did you get your genes data for each taxon? There is a well-known variability (= differences, through their sequences or expressions among others…) in (some) plant genes between individuals of one single taxon (check references with these key-words); from this, at least the scheme of evolution that you propose needs to be more discussed. 1/2 At the end of 3.1, you have to be much more cautious in the eventual (change(s), loss of) gene(s) process; what about accommodation-plasticity, epigenetics, (chromatin modifications, DNA methylation, small RNAs), alleles etc…? (there are many papers about those topics, even academic lessons with examples). 2/ Concerning habitat(s) of these plants, as each plant lives in a given environment and may be influenced (genetically or others), you need the following details 2/1-2. 2/1 As an illustration of the end of your point 3.1, a clear map is necessary to show the distribution of the three taxa (and the location of your plants taken in account, some databases contain the location of their plants…), provide at the same time a table enhancing the whole ecological characteristics of each distribution (temperature, rain, wind, light or shadow, related with altitude-latitude-longitude…, is it what you call “abiotic”?). 2/2 Then you can precise the meaning of harsh /or not harsh habitats: actually “harsh” has no precise ecological meaning at all; moreover “stress” corresponds to a very special concept not applicable automatically for the type of environment you evocate, it is the same for what you call “light stress” or “hormone response”, please precise all these and explain the relationship with the environment(s) of Dendrobium taxa. 3/1 Precise then the functions selected for the genes: as they may have many, how can you decide these functions for Dendrobium taxa compared with other taxa from other botanical families (which function(s) is acting for which plant(s) (all plants are concerned or selected taxa of some families (or other groups)); for instance Arabidopsis is a Brassicaceae (dicotyledonous) and Dendrobium is an Orchidaceae (monocotyledonous), normally it corresponds to very different properties? Moreover, checking briefly in scilit and web of science, there are much more plant taxa related with tcp genes than your report, please check them one by one and see their (putative) functions. 3/2 In order to show (or not, or more or less) the relationship between of habitat and genes-proteins (for the three taxa together? or for each taxon separately?…), your so-called harsh habitat corresponding to some gene plants should be compared with non-harsh habitat with other (?) (parts of) genes…) for the same taxa or closely related. For all these reasons, in the present state of text, you cannot say for instance in 3.2 " we believed that the harsh habitats...", and "The high content of polysaccharides, flavonoids and alkaloids also provide..., and "Therefore, we speculated that TCP genes play a more important role in plant growth and development.", and in the conclusion "resulted in the diversity of biological functions of”.
33- Restrict the results part to only results (values etc.) and remove all sentences which belong actually to the discussion part; for instance the last paragraph of 2.5 belongs actually to the discussion part? in 2.6, "The genes with significantly up-regulated ..." (appearing twice) belong also to the discussion part ? in 2.7, "these results indicated that..." belongs to the discussion part? Moreover, the discussion has to cite some of the precise results instead of general words, it will make the connection between results and discussion parts.
44- More precisely for Dendrobium, references already taken in account by the authors are of interest, however checking briefly in the word of science WOS and scilit (from mdpi) with the key-words of the abstract, other references appear (especially recent ones), and they should be once more selected and used (if relevant…) in order to provide a larger view of this interesting research. Among these are the followings (check also for others…):
[1-14]
1. Feng, S.-G.; Lu, J.-J.; Gao, L.; Liu, J.-J.; Wang, H.-Z. Molecular Phylogeny Analysis and Species Identification of Dendrobium (Orchidaceae) in China. Biochemical Genetics 2013, 52.
2. Garg, R.; Subudhi, P.K.; Varshney, R.K.; Jain, M. Editorial: Abiotic stress: Molecular genetics and genomics, volume II. Frontiers in Plant Science 2023, 13.
3. He, X.; Zhang, W.; Sabir, I.A.; Jiao, C.; Li, G.; Wang, Y.; Zhu, F.; Dai, J.; Liu, L.; Chen, C.; et al. The spatiotemporal profile of Dendrobium huoshanense and functional identification of bHLH genes under exogenous MeJA using comparative transcriptomics and genomics. Frontiers in Plant Science 2023, 14.
4. Huang, H.; Jiao, Y.; Tong, Y.; Wang, Y. Comparative analysis of drought-responsive biochemical and transcriptomic mechanisms in two Dendrobium officinale genotypes. Ind Crop Prod 2023, 199.
5. Jia, X.; Wu, F.; Lu, A.; Tan, D.; Zhang, Q.; He, Y.; Qin, L. Widely Targeted Metabolomics Analysis of Dendrobium officinale at Different Altitudes. Chemistry & Biodiversity 2023, 20.
6. Pan, C.; Chen, S.; Chen, Z.; Li, Y.; Liu, Y.; Zhang, Z.; Xu, Y.; Liu, G.; Yang, K.; Liu, G.; et al. Assessing the geographical distribution of 76 Dendrobium species and impacts of climate change on their potential suitable distribution area in China. Environ Sci Pollut R 2021, 29.
7. Pan, T.; Deng, N.-M.; Guo, W.-X.; Wan, M.-Z.; Zheng, Y.-T.; Chen, S.-Y.; Liu, C.-L.; Li, H.-B.; Liang, S. DnFCA Isoforms Cooperatively Regulate Temperature-Related Flowering in Dendrobium nobile. Biology 2023, 12.
8. Taticharoen, T.; Matsumoto, S.; Chutteang, C.; Srion, K.; Malumpong, C.; Abdullakasim, S. Response and acclimatization of a CAM orchid, Dendrobium Sonia Earsakul to drought, heat, and combined drought and heat stress. Sci Hortic-Amsterdam 2023, 309.
9. Wang, M.T.; Hou, Z.Y.; Li, C.; Yang, J.P.; Niu, Z.T.; Xue, Q.Y.; Liu, W.; Ding, X.Y. Rapid structural evolution of Dendrobium mitogenomes and mito-nuclear phylogeny discordances in Dendrobium (Orchidaceae). Journal of Systematics and Evolution 2022.
10. Wu, L.; Fan, J.; Su, X.; Rao, W.; Duan, Y.; Wang, Y.; Jiang, W.; Sun, Z.; Zhang, L.; Peng, D.; et al. Genome-wide identification of R2R3-MYB Family Genes and their Response to Stress in Dendrobium nobile; 2023.
11. Yang, T.W.; Gao, M.R.; Huang, S.Y.; Zhang, S.W.; Zhang, X.J.; Li, T.; Yu, W.H.; Meng, P.; Shi, Q. GENETIC DIVERSITY AND DNA FINGERPRINTING OF DENDROBIUM OFFICINALE BASED ON ISSR AND SCOT MARKERS. Appl Ecol Env Res 2023, 21.
12. Yang, Y.; Qiu, Y.; Ye, W.; Sun, G.; Li, H. RNA sequencing-based exploration of the effects of far-red light on microRNAs involved in the shade-avoidance response of D. officinale. PeerJ 2023, 11.
13. Zhang, L.; Li, C.; Yang, D.; Wang, Y.; Yang, Y.; Sun, X. Genome-Wide Analysis of the TCP Transcription Factor Genes in Dendrobium catenatum Lindl. Int J Mol Sci 2021, 22.
14. Zhao, Y.; Qin, L.; Tan, D.; Wu, D.; Wu, X.; Fan, Q.; Bai, C.; Yang, J.; Xie, J.; He, Y. Fatty acid metabolites of Dendrobium nobile were positively correlated with representative endophytic fungi at altitude. Frontiers in Microbiology 2023, 14.
5- For all figures put all your letters and numbers in bold as they are almost invisible on my computer-screen. More precisely: for figure 1 indicate each chromosome number (they are indicated in supplementary tables) to be checked and compared rapidly between taxa; for figure 2 clarify in the caption "Different subfamilies" as only two taxonomical families are involved (Orchidaceae and Brassicaceae), I understand that it refers to genes but it is ambiguous; for figure 3, indicate in the caption what represents UTR and CDS, especially for non-specialists; in figure 4 the yellow color on the right side circle is hardly visible; for figure 5 indicate in the caption the meaning of all numbers, moreover the latin names of the taxa on the left side are hardly visible; for figure 6 the letters just below the thick black scales on the left side are absolutely invisible; for figure 7, what does "treat." means ? for figure 8, indicate in the caption the meaning of yellow lines and all other colors; indicate also where the full name of the proteins can be found, especially for non-specialists; for the captions of S1 S2 S3 (or directly on these tiff tables), indicate the abbreviations of amino acids somewhere in full letters to be understood more rapidly; for table 1-5, precise the meaning of ID, Chr (chromosome ?), MW, pl(Da),Tm(℃).
6- Although I am not English-native speaker the English writing needs to be extensively corrected and has to be checked carefully (some expressions like “which named”, “were belonged to class », “While, Motif", “genes were remained”, “genes were response”, “For insurance”, “roles in responded to stress”; the beginnings of 4.1, 4.2 and 4.3 are not understandable in terms of english grammar), see for all other parts of the manuscript.
Minor points are:
1 As I am involved in taxonomy I am very sensible to correct plant taxa names which make their homogeneity and precision at the international level. So put all plants with their latin names in italics and the names of the author(s) at least the first time they appear in the text (from the beginning of the introduction Zea mays, especially in 3.2, see also for all other parts). Use international Plant Names Index (IPNI) https://www.ipni.org/), or equivalent; for instance just for D. nobile, there several varieties and even formis; for cultivars if there is no author’s name(s), put the reference where this name appears firstly in the literature. You also use English names of plants (tomato…), you should write in brackets the latin name plus author(s), see for all other english plant names in your text;
2 The abstract is far too long, restrict it to the key-words of your results and discussion-conclusion; in the details precise what you mean with "detect" as apparently it does not correspond to experiments but to databases analysis?
3 As it is for a journal devoted to plants, insert a figure with the photos of each taxon studied;
4 As the genus Dendrobium contains many taxa quite extensively studied for some of them, precise the reason(s) why you selected "only" three taxa (which I understand due to the work it needs), precise also in the discussion the impact of this selection compared with other taxa not taken in account;
5 At the end of 2.2, precise the reference in brackets;
6 In 2.3, delete one "." after (Figure 3);
7 In 3.1, "Consistent with previous studies..." needs references;
8 Write in full letters CAM and CIN at least once.
It is my point 6: although I am not English-native speaker the English writing needs to be extensively corrected and has to be checked carefully (some expressions like “which named”, “were belonged to class », “While, Motif ", “genes were remained”, “genes were response”, “For insurance”, “roles in responded to stress”; the beginnings of 4.1, 4.2 and 4.3 are not understandable in terms of english grammar).see for all other parts of the manuscript.
Author Response
Dear editor and reviewers,
On behalf of my co-authors, we thank you very much for giving us an opportunity to revise our manuscript, we appreciate editor and reviewers very much for their positive and constructive comments and suggestions. Those comments are all valuable and very helpful for revising and improving our manuscript. We have studied comments carefully and tried our best to revise our manuscript according to the comments, which we hope meet with approval.
Response to Reviewer 3:
- The first important point is that it is very ambiguous to read this manuscript, as we do not know exactly if results correspond to database or to laboratory experiment (your point 4.7? For instance in 2.6, "were obviously expressed" means with database or experimentally? See also in 3.1 “members were identified”; in 4.5, do you mean that you experimented for "The genomic DNA sequences of 1500 bp upstream of TCP genes were "extracted"...", if so how did you get the plants and how did you cultivate them and where does it appears in material and methods?
Response:
Many thanks for your valuable and helpful suggestion.
As can be seen from Figure 7A, among the 24 TCP genes of Dendrobium officinale, 8 DoTCPs (DoTCP6, DoTCP9, DoTCP11, DoTCP13, DoTCP14, DoTCP15, DoTCP19 and DoTCP21) were highly expressed. "The genomic DNA sequences of 1500 bp upstream of TCP genes were " has been modified to “The upstream 1500bp of TCP gene translation initiation site was extracted as the presumed promoter ”
- The second important point concerns the discussion and the (complex but very interesting) hypotheses enhanced, unfortunately too general and too weakly sustained because of 1/genes, 2/habitats and 3/relationship genes-proteins-habitats. 1/1 From how many entries (= number of plants analysed; from databases or others) did you get your genes data for each taxon? There is a well-known variability (= differences, through their sequences or expressions among others…) in (some) plant genes between individuals of one single taxon (check references with these key-words); from this, at least the scheme of evolution that you propose needs to be more discussed. 1/2 At the end of 3.1, you have to be much more cautious in the eventual (change(s), loss of) gene(s) process; what about accommodation-plasticity, epigenetics, (chromatin modifications, DNA methylation, small RNAs), alleles etc…? (there are many papers about those topics, even academic lessons with examples). 2/ Concerning habitat(s) of these plants, as each plant lives in a given environment and may be influenced (genetically or others), you need the following details 2/1-2. 2/1 As an illustration of the end of your point 3.1, a clear map is necessary to show the distribution of the three taxa (and the location of your plants taken in account, some databases contain the location of their plants…), provide at the same time a table enhancing the whole ecological characteristics of each distribution (temperature, rain, wind, light or shadow, related with altitude-latitude-longitude…, is it what you call “abiotic”?). 2/2 Then you can precise the meaning of harsh /or not harsh habitats: actually “harsh” has no precise ecological meaning at all; moreover “stress” corresponds to a very special concept not applicable automatically for the type of environment you evocate, it is the same for what you call “light stress” or “hormone response”, please precise all these and explain the relationship with the environment(s) of Dendrobium taxa. 3/1 Precise then the functions selected for the genes: as they may have many, how can you decide these functions for Dendrobium taxa compared with other taxa from other botanical families (which function(s) is acting for which plant(s) (all plants are concerned or selected taxa of some families (or other groups)); for instance Arabidopsis is a Brassicaceae (dicotyledonous) and Dendrobium is an Orchidaceae (monocotyledonous), normally it corresponds to very different properties? Moreover, checking briefly in scilit and web of science, there are much more plant taxa related with tcp genes than your report, please check them one by one and see their (putative) functions. 3/2 In order to show (or not, or more or less) the relationship between of habitat and genes-proteins (for the three taxa together? or for each taxon separately?…), your so-called harsh habitat corresponding to some gene plants should be compared with non-harsh habitat with other (?) (parts of) genes…) for the same taxa or closely related. For all these reasons, in the present state of text, you cannot say for instance in 3.2 " we believed that the harsh habitats...", and "The high content of polysaccharides, flavonoids and alkaloids also provide..., and "Therefore, we speculated that TCP genes play a more important role in plant growth and development.", and in the conclusion "resulted in the diversity of biological functions of”.
Response:
Many thanks for your valuable and helpful suggestion.
The question you raised is very correct and meaningful. This problem is explained in another article of our research group. The article was under submission at the time, so it was confidential. This article is now available and referenced in this article. “39. Xue, Q.; Yang, J.; Yu, W.; Wang, H.; Hou, Z.; Li, C.; Xue, Q.; Liu, W.; Ding, X.; Niu, Z. The climate changes promoted the chloroplast genomic evolution of Dendrobium orchids among multiple photosynthetic pathways. BMC Plant Biol. 2023 10;23(1):189.”
- Restrict the results part to only results (values etc.) and remove all sentences which belong actually to the discussion part; for instance the last paragraph of 2.5 belongs actually to the discussion part? in 2.6, "The genes with significantly up-regulated ..." (appearing twice) belong also to the discussion part ? in 2.7, "these results indicated that..." belongs to the discussion part? Moreover, the discussion has to cite some of the precise results instead of general words, it will make the connection between results and discussion parts.
Response:
Many thanks for your valuable and helpful suggestion.
The results section has been modified so that the results section only describes the research results and the discussion section has been modified
- More precisely for Dendrobium, references already taken in account by the authors are of interest, however checking briefly in the word of science WOS and scilit (from mdpi) with the key-words of the abstract, other references appear (especially recent ones), and they should be once more selected and used (if relevant…) in order to provide a larger view of this interesting research. Among these are the followings (check also for others…):
Response:
Many thanks for your valuable and helpful suggestion.
The references have been added and modified.
- For all figures put all your letters and numbers in bold as they are almost invisible on my computer-screen. More precisely: for figure 1 indicate each chromosome number (they are indicated in supplementary tables) to be checked and compared rapidly between taxa; for figure 2 clarify in the caption "Different subfamilies" as only two taxonomical families are involved (Orchidaceae and Brassicaceae), I understand that it refers to genes but it is ambiguous; for figure 3, indicate in the caption what represents UTR and CDS, especially for non-specialists; in figure 4 the yellow color on the right side circle is hardly visible; for figure 5 indicate in the caption the meaning of all numbers, moreover the latin names of the taxa on the left side are hardly visible; for figure 6 the letters just below the thick black scales on the left side are absolutely invisible; for figure 7, what does "treat." means ? for figure 8, indicate in the caption the meaning of yellow lines and all other colors; indicate also where the full name of the proteins can be found, especially for non-specialists; for the captions of S1 S2 S3 (or directly on these tiff tables), indicate the abbreviations of amino acids somewhere in full letters to be understood more rapidly; for table 1-5, precise the meaning of ID, Chr (chromosome ?), MW, pl(Da),Tm(℃).
Response:
Many thanks for your valuable and helpful suggestion.
In order to improve the clarity of the pictures, the text in the pictures has been enlarged, and the clarity of the pictures has been improved, making the pictures easier to read. At the same time, some content in the figure has been modified. In the supplementary tables, the abbreviation has been added to the full name.
- Although I am not English-native speaker the English writing needs to be extensively corrected and has to be checked carefully (some expressions like “which named”, “were belonged to class », “While, Motif", “genes were remained”, “genes were response”, “For insurance”, “roles in responded to stress”; the beginnings of 4.1, 4.2 and 4.3 are not understandable in terms of english grammar), see for all other parts of the manuscript.
Response:
Many thanks for your valuable and helpful suggestion.
The writing of material method has been modified to make the language more logical.
- As I am involved in taxonomy I am very sensible to correct plant taxa names which make their homogeneity and precision at the international level. So put all plants with their latin names in italics and the names of the author(s) at least the first time they appear in the text (from the beginning of the introduction Zea mays, especially in 3.2, see also for all other parts). Use international Plant Names Index (IPNI) https://www.ipni.org/), or equivalent; for instance just for D. nobile, there several varieties and even formis; for cultivars if there is no author’s name(s), put the reference where this name appears firstly in the literature. You also use English names of plants (tomato…), you should write in brackets the latin name plus author(s), see for all other english plant names in your text;
The abstract is far too long, restrict it to the key-words of your results and discussion-conclusion; in the details precise what you mean with "detect" as apparently it does not correspond to experiments but to databases analysis?
As it is for a journal devoted to plants, insert a figure with the photos of each taxon studied;
As the genus Dendrobium contains many taxa quite extensively studied for some of them, precise the reason(s) why you selected "only" three taxa (which I understand due to the work it needs), precise also in the discussion the impact of this selection compared with other taxa not taken in account;
At the end of 2.2, precise the reference in brackets;
In 2.3, delete one "." after (Figure 3);
In 3.1, "Consistent with previous studies..." needs references;
Write in full letters CAM and CIN at least once.
Response:
Many thanks for your valuable and helpful suggestion.
The content of the abstract has been streamlined to make it more organized.
The full names of CAM and CIN have been supplemented. The Latin name has been modified.
The quality of genomic data splicing of Dendrobium was not good enough or there was no annotation information, so these three Dendrobium genomes were selected for analysis and these D. officinale, D. nobile, and D. chrysotoxum are important to the Dendrobium.
The writing of the full text has been checked and revised.
Round 2
Reviewer 2 Report
I could not see any significant improvement.
And most surprisingly, authors removed some of the suggestions in place of responding.
Horrible.
Author Response
Dear editor and reviewers,
We would like to express our sincere gratitude for granting us the opportunity to revise our manuscript. On behalf of my co-authors, we deeply appreciate the editor and reviewers for their invaluable and constructive comments and suggestions. These remarks have proved to be immensely valuable in enhancing the quality of our manuscript. We have meticulously analyzed the comments and have made every effort to incorporate them into our revisions, aiming to ensure their approval.
Response to Reviewer 2:
- For instance line 28, play important roles in responded to stress. ?
Response:
Many thanks for your valuable and helpful suggestion.
The original text has been amended to read “Using qRT-PCR to determine their expression patterns under MeJA treatment, 4 DoTCPs (DoTCP2, DoTCP4, DoTCP6, and DoTCP14) were significantly upregulated under MeJA treatment, which indicates that the TCP genes may play important roles in responding to stress resistance. ”
- How many TCP genes are present in Arabidopsis? Why number is variable in different species of Dendrobium? It seems to be not identified properly.
Response:
Many thanks for your valuable and helpful suggestion.
The number of TCP genes in Arabidopsis thaliana is 24, as shown in 2.1.(To identify TCP family genes in D. officinale, D. nobile, and D. chrysotoxum, we used the BLAST method and hidden Markov model (HMM) to determine TCPs from their genome sequences, using 24 known TCP protein sequences from Arabidopsis as references.)
The reasons why the number of different Dendrobium species is variable are presented in the discussion. The evolution of TCP genes in Dendrobium was discussed with these three Dendrobium species (D. officinale, D. nobile, and D. chrysotoxum)as research objects, and an important point was grasped: TCP genes in CIN branch were lost in D. chrysotoxum, and the reason for this phenomenon was analyzed.
- 2. Phylogenetic relationship and gene structure 365
“Use MAFFT v7.487 software to compare the full-length amino acid sequences of TCP 366 [45], and use RAxML v1.3 software with a bootstrap value of 1,000 to build the maximum 367 likelihood phylogenetic trees.”
Is it the way of writing? The methods should be completely revised. Authors may take the help for language correction.
Response:
Many thanks for your valuable and helpful suggestion.
The writing of material method has been modified to make the language more logical.
The original text has been amended to read “The full-length amino acid sequences of TCP [45] were compared using MAFFT v7.487 software and the maximum likelihood phylogenetic trees were constructed using RAxML v1.3 software with a bootstrap value of 1,000.”
- I could not see any internal control in Rt PCR, usually two gene are now used as described mdpi.com/2075-1729/12/7/941.
Response:
Many thanks for your valuable and helpful suggestion.
The gene GAPDH was used as the internal reference gene. Primer sequences are presented in Table S5.
- Authors should also include evolutionary analysis as described www.mdpi.com/2223-7747/11/7/911, because this study is mostly in-silico.
Response:
Many thanks for your valuable and helpful suggestion.
Part of the evolutionary analysis is included in this paper, as shown in result 2.2. “To explore the relationship of Dendrobium TCP genes, we performed phylogenetic analysis of 84 genes from D. officinale, D. nobile, D. chrysotoxum, and A. thaliana (Figure 2). The results inferred from the NJ tree have shown that the TCP genes were classified into two main classical subfamilies (class â… and class â…¡). Among them, 28 genes (9 DoTCPs, 10 DnTCPs, and 9 DcTCPs) and 33 genes (15 DoTCPs, 13 DnTCPs, and 5 DcTCPs) belonged to class I and class â…¡, respectively. (Figure 2). Class II was classified into two subclasses: CYC/TB1 (4 DoTCPs, 1 DnTCPs, and 2 DcTCPs) and CIN (11 DoTCPs, 12 DnTCPs, and 3 DcTCPs) (Figure 2). ”
- RT PCR methods and interaction methods should be separately written. Authors should also use stitch for interaction analysis. You may follow mdpi.com/2223-7747/11/5/587
Response:
Many thanks for your valuable and helpful suggestion.
The experimental methods for Quantitative real-time PCR and Protein-protein interaction network have been described separately in Materials and Methods 4.7 and 4.8. Both the software tools STITCH and STRING are capable of analyzing protein interactions, with STITCH additionally capable of analyzing Chemical-Protein interaction networks. As the analysis in this paper does not involve chemistry, STRING has been selected for the analysis.
- Figure quality should be improved. Most of the written parts are invisible in the majority of figures.
In Figure 7A, I don’t see any expression value in most of the genes. How it is possible?
Response:
Many thanks for your valuable and helpful suggestion.
In order to improve the clarity of the pictures, the text in the pictures has been enlarged, and improving the clarity of the pictures and making them easier to read.
In Figure 7A, the expression values of genes have been labeled in the figure.
- Discussion seems to be repetition of results in the many paragraphs. Authors may refer the suggested Ms during the revision of Ms. Only important findings could be discussed.
Response:
Many thanks for your valuable and helpful suggestion.
The content of the discussion section has been modified. Furthermore, the content of some results has been deleted, and the key content has been retained for discussion.
- Conclusion should be rewritten with future perspective. It should not be the repeat of abstract.
Response:
Many thanks for your valuable and helpful suggestion.
The conclusion has been revised and written.
Round 3
Reviewer 2 Report
Ms lacks several basic things including statistical analysis methods and parameters. And many more things were earlier suggested has not been upto the mark. Why GAPDH was used as internal control?
How most of the genes have 0 expression value, that raises concern about the analysis method.
No comment